# Alterations in the Microbiome of Horses Affected with Fecal Water Syndrome

**DOI:** 10.3390/vetsci12080724

**Published:** 2025-07-31

**Authors:** Madison M. Porter, Daniel J. Davis, Zachary L. McAdams, Kile S. Townsend, Lynn M. Martin, Christopher Wilhite, Philip J. Johnson, Aaron C. Ericsson

**Affiliations:** 1College of Veterinary Medicine (CVM), University of Missouri (MU), 1520 E. Rollins Drive, Columbia, MO 65211, USA; mpfhb@missouri.edu (M.M.P.); davisdaniel@missouri.edu (D.J.D.); zlmg2b@missouri.edu (Z.L.M.); townsendks@missouri.edu (K.S.T.); lynn.martin@missouri.edu (L.M.M.); 2Animal Modeling Core, University of Missouri (MU), 4011 Discovery Drive, Columbia, MO 65201, USA; 3Metagenomics Center (MUMC), University of Missouri, 4011 Discovery Drive, Columbia, MO 65201, USA; 4Department of Veterinary Medicine and Surgery, MU College of Veterinary Medicine, 900 E. Campus Drive, Columbia, MO 65211, USA; 5Wilhite and Frees, 21215 S. Peculiar Drive, P.O. Box 425, Peculiar, MO 64078, USA; clwdvm@gmail.com

**Keywords:** equine, fecal water syndrome, microbiome, dysbiosis

## Abstract

Fecal water syndrome is a condition in horses wherein normal feces are followed by liquid feces. While the exact cause is unknown, it may result from changes in the native gut bacteria. Here, fecal samples were collected from horses affected with fecal water syndrome and unaffected horses on the same farms. Laboratory methods were used to identify bacteria present in affected and unaffected horses. Analysis of the data revealed characteristic changes in the fecal bacteria of horses affected with fecal water syndrome, including the increased abundance of certain bacteria such as *Alloprevotella* species.

## 1. Introduction

Fecal water syndrome (FWS), or ‘free fecal liquid’, is a recently observed condition of horses, characterized by two-phase separation of feces, one solid and one liquid phase, voided together or separately, that is being increasingly recognized [1,2]. The condition leads to soiling and irritation of the hindquarters as well as being unhygienic and aesthetically displeasing. Owners of FWS-affected equids regard the condition as an impediment to riding and training. Although the prevalence of FWS is largely unknown, it has been shown to be present in 9% of Swiss Warmbloods [3]. Several potential causative factors have been previously investigated, including the type of hay fed, the horse’s social status, and bacterial dysbiosis of the gastrointestinal tract [1,4].

The bacterial microbiome of the equine gut is a rich and dynamic ecosystem sustained by regular intake of dietary substrates. During disease processes affecting colonic function, the richness of the fecal microbiome is reduced [5]. The relationship is symbiotic, as the bacteria produce a wide range of metabolites and small molecules, which are absorbed into peripheral circulation via the portal vein or act locally on intestinal epithelial cells and colonocytes to influence motility and other functions [6,7].

Previous studies comparing the fecal microbiomes of horses with and without FWS have produced varying results. While three initial studies unanimously failed to detect differences in overall alpha or beta diversity between FWS-affected and control horses [8,9,10], the study benefitting from the highest sample size (*n* = 50/group) and a longitudinal study design found significant differences in the relative abundances (RAs) of *Alloprevotella* and *Bacillus* spp. [10] A subsequent study similarly failed to detect significant effects of FWS on alpha or beta diversity or the RAs of *Alloprevotella* and *Bacillus* spp. [11] However, the mean RA of the *Alloprevotella* sp. was 55% greater in FWS-affected horses than control horses (*p* = 0.202) [11], demonstrating the same directionality as the differential abundance reported by others [10].

The aforementioned studies were performed in Canada or Europe, and changes in the microbiome associated with FWS have not been investigated in the United States. The primary aims of the current study were to evaluate and compare the fecal microbiomes of horses affected with FWS and unaffected control horses from the same farms in the central U.S. and to identify FWS-associated features in the fecal microbiome that could be used as diagnostic or therapeutic targets or reveal information about the underlying process.

## 2. Materials and Methods

### 2.1. Horses

Feces were obtained from adult horses affected with FWS. For the purpose of this study, diagnosis of FWS was based on the passage of feces in two distinct phases (confirmed through veterinary supervision), one being liquid and the other solid. The veterinary diagnosis of FWS has been defined as the passage of normal manure, attended (before, during, or after defecation) by watery liquid running freely from the anus [12]. Horses with consistently soft or wet feces (diarrhea) were not included. Recruited horses were regarded as broadly healthy in other respects, based on the results of physical examination performed by the attending local veterinarian. Specific information regarding each horse’s signalment, overall health (independent of FWS), and management was obtained: age, breed, gender, body condition score, ration (pasture, hay, grain, or a combination), purpose, current medications, duration of signs of FWS, and parasite management.

### 2.2. Fecal Sample Collection

Freshly voided fecal samples were obtained either by the attending veterinarian or the owner. Fecal boli were collected directly from the ground using latex gloves, taking care to prevent ground surface contamination. If available, fecal samples from one or two paired control (healthy adult) horses at the same barn were similarly collected. Samples were promptly placed in air-tight plastic containers, frozen, and transported to the laboratory on ice. Frozen samples were maintained at −80 °C for processing as a batch.

### 2.3. DNA Extraction

Extraction of fecal DNA was performed with QIAamp PowerFecal Pro DNA kits (Qiagen, Hilden, Germany) following manufacturer instructions, except that the samples were homogenized in the bead tubes with a TissueLyser II (Qiagen, Hilden, Germany) for 10 min instead of the vortex adapter included in the manufacturer protocol. DNA was eluted in 100 μL of EB buffer (Qiagen, Hilden, Germany). Extraction yields were assessed using fluorometry (Qubit 2.0, Invitrogen, Waltham, MA, USA) with Quant-iT BR dsDNA reagent kits (Invitrogen, Waltham, MA, USA). The DNA concentrations were then normalized for library preparation. The DNA was then used to generate 16S rRNA amplicon libraries that were sequenced using the Illumina MiSeq platform.

### 2.4. 16S rRNA Library Preparation and Sequencing

Amplification and sequencing of libraries were performed at the University of Missouri (MU) Genomics Technology Core. Microbial 16S rRNA amplicon libraries were generated via amplification of the V4 region of the 16S rRNA gene using universal dual-indexed forward and reverse primers (U515F/806R) [13] flanked by standard Illumina adapter sequences. Polymerase chain reactions were performed in 50 μL reactions comprising 100 ng metagenomic DNA, primers (0.2 μM each), dNTPs (200 μM each), and Phusion high-fidelity DNA polymerase (1U, Thermo Fisher). Amplification parameters were 98 °C (3 min) + [98 °C (15 s) + 50 °C (30 s) + 72 °C (30 s)], 25 cycles + 72 °C (7 min). Libraries were combined, mixed, and purified using Axygen Axyprep MagPCR clean-up beads. Products were washed several times with 80% ethanol, and the dried pellet was resuspended in EB buffer (Qiagen, Hilden, Germany), incubated at room temperature for 2 min, and then placed on a magnetic stand for 5 min. The final amplicon pool was assessed using a Fragment Analyzer automated electrophoresis system (Advanced Analytical), quantified using quant-iT HS dsDNA reagent kits, and diluted according to the Illumina standard protocol for sequencing as 2 × 250 base pair (bp) paired-end reads on a MiSeq instrument.

### 2.5. Bioinformatics

The 16S rRNA sequences were processed using Quantitative Insights Into Microbial Ecology 2 (QIIME2) v2021.8.26 [14]. Cutadapt [15] was used to remove Illumina adapters and primer sequences from forward and reverse reads. DADA2 [16] was used to truncate reads to 150 bp and establish unique amplicon sequence variants (ASVs). Taxonomies were assigned to sequences using an a sklearn algorithm [17] and then the SILVA29 v138 reference database [18]. Alpha diversity metrics (Chao-1 and Simpson indices) were calculated using the microbiome [19] and vegan [20,21] libraries. Principal coordinate analysis (PCoA) was performed using both Bray–Curtis (weighted) and Jaccard (unweighted) distances to visualize beta diversity. Distance matrices were made using the vegdist function in the vegan library and a quarter-root transformed feature table. PCoA was performed using the ape library [22]. The cladogram was made using Graphlan v1.1.4 [23].

### 2.6. Statistics

Univariate data (reported as means ± SE) first were tested for normality using the Shapiro–Wilk method, followed by the appropriate parametric or non-parametric test. Whenever possible and practical, multifactor tests (e.g., two-way analysis of variance [ANOVA]) were used, with day and horse as factors. Multivariate statistical testing was based on permutational multivariate ANOVA (PERMANOVA) performed with 9999 permutations. Like PCoA, PERMANOVA was performed using Jaccard and Bray–Curtis distances. Differences in microbiome richness and alpha and beta diversity were determined using traditional univariate (Student’s *t*-test) or multivariate (PERMANOVA) statistics, as appropriate, and visualized using box plots and principal coordinate analysis. Differential abundance testing at all taxonomic levels was performed using ANCOM-BC2 [24], ALDEx2 [25], and linear discriminant analysis effect size (LEfSe) [26] analysis. Significance was defined as *p* < 0.05 for traditional univariate and multivariate tests; *p* values adjusted for multiple tests, (*p*_adj_) < 0.05, were considered significant following differential abundance testing. The minimum group sample size was set at 32, using group mean and sigma values for the *Alloprevotella* sp. approximated from Lindroth et al. and α = 0.05 to provide a power (1 − β) of 0.85.

## 3. Results

### 3.1. Study Population

This study included 32 horses affected with FWS and 51 healthy control horses. The average age of the affected horses was 20 years (range, 5–30 years), and the average age of the control horses was 17 years (range, 6–31 years). The average duration of FWS was 3 years (range, 0.5–10 years). Of the affected horses, 75% (24/32) were geldings and the remainder (8/32) were mares. There were 16 breeds represented in the affected group, and the reported uses of the horses included pleasure riding, dressage, jumper, broodmare, trail riding, and retirement. Fifty-two percent of the affected horses were receiving at least one medication at the time samples were obtained. Firocoxib and pergolide mesylate were the most commonly prescribed medications in both control and case horses. The control group comprised 20 different breeds, and the reported uses of the horses were similar to the affected horses. A complete list of collected information can be found in Appendix A.

Fecal samples were collected from 32 symptomatic horses and 51 control horses. The first round of samples was collected between February and April of 2022, and a second round was collected between August and October of the same year.

### 3.2. Bacterial Microbiome Analysis

First, richness and alpha diversity were estimated using the Chao1 and Shannon indices, respectively. Testing via the Wilcoxon rank-sum test failed to detect a significant difference in richness (*p* = 0.775, Figure 1A) or alpha diversity (*p* = 0.126, Figure 1B).

Next, beta diversity was assessed using both weighted (Bray–Curtis) and unweighted (Jaccard) dissimilarities. Principal coordinate analysis (PCoA) revealed minimal FWS-associated separation of samples based on Bray–Curtis (Figure 1A) or Jaccard (Figure 1B) dissimilarities. However, one-way permutational multivariate analysis of variance (PERMANOVA) detected a significant difference based on Bray–Curtis (*p* = 0.0018, F = 1.81) and Jaccard (*p* = 7 × 10^−4^, F = 1.51) dissimilarities.

Recognizing that other factors may influence the risk of developing FWS or the composition of the fecal microbiome, we first tested for differences between FWS-affected and control horses in rations consumed, current medications, and age. Chi-square tests comparing access to hay (χ2 = 0.036, df = 1, *p* = 0.85) or pastures (χ2 = 0.005, df = 1, *p* = 0.945) both failed to detect significant associations with FWS, and there was no significant interaction (hay × pasture, χ2 = 1.51, df = 2, *p* = 0.471). Grain was consumed by all horses. Similar results were obtained with current medications. Notably, a significant difference was detected in the ages of affected and control horses (*p* = 0.030, *t*-test), suggesting age should be considered as a covariate in beta diversity. Based on clustering observed in PCoA (Figure 2C,D), the immediate environment of the farm represents an additional covariate. In order to determine the variance and effect associated with the farm before assigning the variances and effects of FWS status and age, PERMANOVA was repeated using the formula [dist ~ farm + (group × age)]. Using Bray–Curtis dissimilarities, the farm (*p* < 0.001, F = 1.68) and group (*p* < 0.001, F = 2.38) were both identified as significant factors, while age (*p* = 0.071, F = 1.10) and group × age interactions (*p* = 0.470, F = 1.01) were not significant. Similar effects of the farm (*p* < 0.001, F = 1.39) and group (*p* < 0.001, F = 1.77) were detected using Jaccard dissimilarities. These outcomes suggest a difference between affected and control horses, with an effect size equivalent to or greater than the inter-farm variability.

Across the study population, the core microbiome at the phylum and family levels aligned with prior studies. Nineteen families were detected at greater than 1% relative abundance (RA) in at least one individual (Appendix A). Differential abundance testing using relatively conservative methods including ANCOM-BC2 and ALDEx2 yielded zero significantly different taxonomic features between groups based on health status. Using a slightly less stringent approach, linear discriminant analysis (LDA) effect size (LEfSe) analysis identified 49 taxonomic markers associated with either FWS or control health status (Appendix A). The five and seven families enriched in the control and FWS-affected horses, respectively, both included members of *Actinomycetota*, *Bacillota*, and *Bacteroidota*, as well as other group-specific taxa (Figure 3). Among the specific genera enriched in the control horses, the lowest *p* values were associated with members of *Bacteroidota*, including the dgA-11 gut group (*p*_adj_ = 3.04 × 10^−4^) and an unresolved member of order *Bacteroidales* (*p*_adj_ = 0.0024), and Clostridial organisms within the phylum *Bacillota*, including *Eubacterium* (*p*_adj_ = 0.0015) and *Pygmaiobacter* (*p*_adj_ = 0.0012), among others.

In contrast, among the genera enriched in FWS-affected samples, the lowest *p* values were associated with *Alloprevotella* (*p*_adj_ = 8.51×10^−4^), other Clostridial organisms including the *Christensenellaceae* R-7 group (*p*_adj_ = 0.005) and *Acetitomaculum* (*p*_adj_ = 0.0058), and an unresolved member of *Rhodobacteraceae* (*p*_adj_ = 0.0068).

## 4. Discussion

Multiple prior studies have investigated differences in the equine microbiome associated with FWS in Canada and Europe [8,9,10,11]. While none identified significant differences between affected and control horses in alpha or beta diversity, two of them identified taxa enriched in affected or control horses using LEfSe [8] or Wilcoxon signed-rank tests [10]. There was, however, little agreement between prior studies in the presence or identity of taxa enriched in samples from FWS or control horses. In well-powered work from Lindroth et al. (*n* = 50 horses per group) [10], two bacterial taxa demonstrated differential abundance across all three sampling time-points. The *Bacillus* sp. was enriched in control horses, while the *Alloprevotella* sp. was enriched in FWS. Here, two members of class *Bacilli* (*Erysipelatoclostridiaceae* UCG-004 and *Mycoplasma* sp.) were enriched in samples from control horses, and the *Alloprevotella* sp. was enriched in samples from FWS-affected horses. The current data thus, at least partially, agree with their findings. *Alloprevotella* sp. is a normal member of the healthy equine microbiome, identified in other studies as a feature associated with old age or obesity [27] and changes in feed [28]. Similarly, we speculate that the *Alloprevotella* sp. represents an indicator or biomarker of FWS rather than a contributing factor, although this is unknown.

The discrepancies between studies may reflect the combined result of confounding variables such as geographic region [29], time of year, horse breed, management strategies, and other factors [27]. The present data agree with prior work showing site-specific effects [9], although no interaction was detected between farms and the effect of health status (i.e., FWS) on beta diversity. Discrepancies between this and prior studies of FWS may also reflect differences in technical aspects of 16S rRNA library preparation, sequencing, and informatics. Prior work from other groups [8,9,10,11] used different combinations of four different primer pairs (targeting three different gene regions), three different sequencers, and four different types of annotation software using two different databases. Lastly, insufficient sample size may have contributed to earlier negative findings, as three of the aforementioned studies relied on 16 or fewer FWS cases. While our findings agree with those of Lindroth et al. [10], the effect size was modest in both studies, suggesting the need for a large sample size to identify these effects. Future studies investigating the influence of the microbiome in FWS will thus need to be sufficiently powered.

One limitation of the current study was the inability to identify or consider protozoal organisms in the gut microbiome. While protozoal organisms like *Cryptosporidium* and *Giardia* are typically associated with diarrhea in foals [30,31], reports of *C. andersoni*-associated diarrhea in a group of adult horses suggest certain subtypes may be pathogenic in adults [32]. Other protozoal organisms capable of infecting horses, including *Balantidium coli* and *Eimeria leuckarti*, are considered incidental findings [33,34] but have been suspected to contribute to equine diarrhea [34]. Future studies employing other ribosomal gene targets could be used to survey protozoal and fungal members of the microbiome. Additionally, rations were treated as binary covariates in analyses assessing their relationships with FWS. More detailed analyses incorporating the proportions of each ration may be more informative. However, as horses on the same farm frequently receive similar rations, we speculate that differences in rations between farms likely contribute to the observed effect of farms on microbial beta diversity. Additional studies with large sample sizes are needed to accurately assign hazard ratios to dietary components and other possible risk factors (e.g., animal use, temperament, climate).

## 5. Conclusions

In conclusion, FWS was associated with differences in beta diversity and the relative abundance of certain taxa in feces. Notably, the association between FWS and fecal relative abundance of *Alloprevotella* align with previous findings from other groups, suggesting proliferation of this genus may represent a hallmark change in the fecal microbiome of horses affected with FWS. Future work is needed to elucidate the etiology of the condition.

## Figures and Tables

**Figure 1 vetsci-12-00724-f001:**
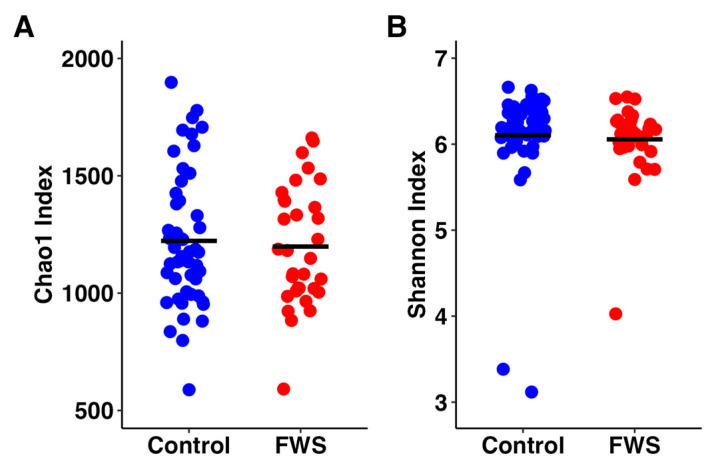
Dot plots showing the Chao1 index (**A**) and Shannon index (**B**) as estimates of richness and alpha diversity, respectively, in unaffected control horses (*n* = 51) and horses affected with fecal water syndrome (FWS, *n* = 32). Bars indicate group means.

**Figure 2 vetsci-12-00724-f002:**
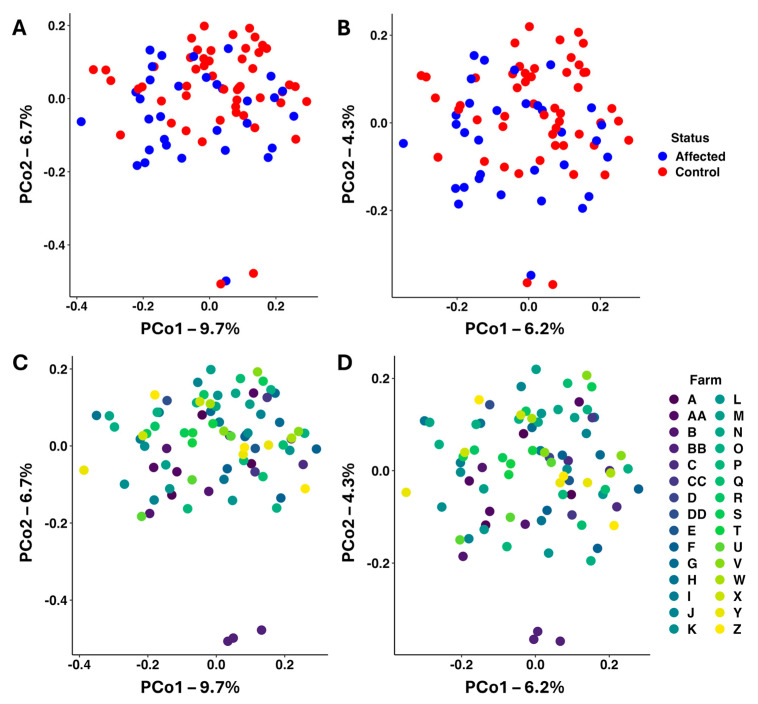
Principal coordinate analysis plots showing beta diversity among fecal samples collected from unaffected control horses (*n* = 51) and horses affected with fecal water syndrome (FWS, *n* = 32), colored according to presence or absence of FWS (**A**,**B**) or farm (**C**,**D**) and based on weighted (**A**,**C**) and unweighted (**B**,**D**) dissimilarities.

**Figure 3 vetsci-12-00724-f003:**
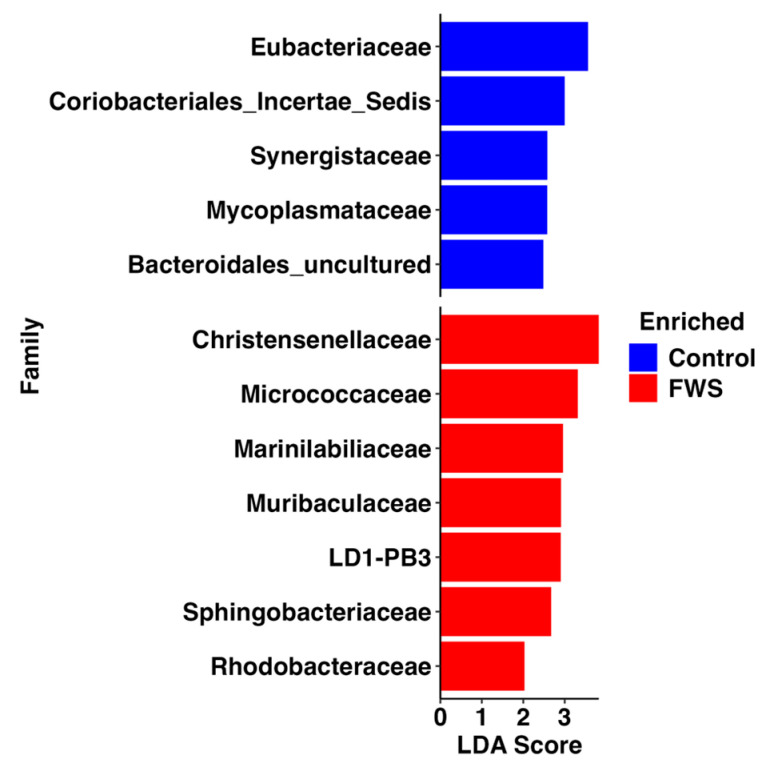
Bar chart showing taxonomic markers at the level of the genus associated with unaffected control horses (*n* = 51) or horses affected with fecal water syndrome (FWS, *n* = 32) based on linear discriminant analysis (LDA) effect size analysis (LEfSe).

## Data Availability

All 16S rRNA sequencing data supporting the current work are freely available at the National Center for Biotechnology Information (NCBI) Sequence Read Archive (SRA) as BioProject PRJNA1032075.

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
