# Peer review of "Alterations in the Microbiome of Horses Affected with Fecal Water Syndrome"

_vetsci, 2025, doi:10.3390/vetsci12080724_

Round 1

Reviewer 1 Report

Comments and Suggestions for Authors

Dear authors,

Thank you for submitting this manuscript.

While the central idea of the manuscript appears promising, the way the results are explored seems insufficiently supported.

Introduction:
One would expect the introduction to clarify why Balantidium coli is given such importance. However, the entire manuscript seems to suggest that this organism should not hold a central role in the study — whether in terms of objectives, methodology, or results. There is no evidence of high prevalence in diseased horses, nor is there strong evidence of its presence in other species. As mentioned by the authors in line 63, the horse is an uncommon host, and the line 64 indicates that there is little information about this organism in equines. As such, the reason for placing Balantidium coli at the center of the objective, results, and especially the title of the manuscript remains unexplained.

The mere observation, as mentioned in lines 68–71, that “our hospital have suggested that motile ciliated protozoa, which are morphologically consistent with B. coli, are often observed in feces from horses affected with fecal water syndrome”, is not sufficient to justify placing this organism at the center of the study.

I strongly recommend that the focus of this study be on identifying differences in the microbiota between horses with and without fecal water syndrome (FWS). This would provide clinically relevant insights and could assist in the selection of appropriate treatments.

The results themselves appear to support this interpretation, as Balantidium coli showed no particular relevance — as expected from the existing literature.

I also suggest modifying the manuscript title to:
"Alterations in the Microbiota of Horses with FWS"

Finally, the aim of the study is not clearly defined in the introduction. It is important that one of the final lines of the introduction explicitly states: “The aim of this work was…”.

Materials and methods

Line 76-77: it is necessary an reference to validate this method of diagnosis.

2.7) Statistics: you need to indicate p value of significance

Conclusions:

A conclusion stating that B. coli is not the causative agent of FWS, without providing justification for why this is an important conclusion, is a conclusion based on unfounded reasoning. The entire introduction suggested that this bacterium would not play a significant role in this pathology.

Author Response

Reviewer 1

Dear authors,

Thank you for submitting this manuscript.

While the central idea of the manuscript appears promising, the way the results are explored seems insufficiently supported.

Introduction
Comment 1: One would expect the introduction to clarify why Balantidium coli is given such importance. However, the entire manuscript seems to suggest that this organism should not hold a central role in the study — whether in terms of objectives, methodology, or results. There is no evidence of high prevalence in diseased horses, nor is there strong evidence of its presence in other species. As mentioned by the authors in line 63, the horse is an uncommon host, and the line 64 indicates that there is little information about this organism in equines. As such, the reason for placing Balantidium coli at the center of the objective, results, and especially the title of the manuscript remains unexplained.

The mere observation, as mentioned in lines 68–71, that “our hospital have suggested that motile ciliated protozoa, which are morphologically consistent with B. coli, are often observed in feces from horses affected with fecal water syndrome”, is not sufficient to justify placing this organism at the center of the study.

I strongly recommend that the focus of this study be on identifying differences in the microbiota between horses with and without fecal water syndrome (FWS). This would provide clinically relevant insights and could assist in the selection of appropriate treatments.

The results themselves appear to support this interpretation, as Balantidium coli showed no particular relevance — as expected from the existing literature.

Response 1: We sincerely appreciate the Reviewer’s assessment of our work and understand their concerns with the focus on Balantidium coli. Our interest in B. coli in the context of FWS extends from a clinical case of FWS in which B. coli was identified.  That being said, concerns from this and other Reviewers regarding the B. coli data cannot be adequately addressed through revision and we have opted to remove those data entirely from this manuscript and focus solely on the changes in the microbiota, as recommended by the Reviewer. As this resulted in the removal of a substantial portion of the Introduction and Discussion, the remaining text in those sections has been expanded slightly (lines 44-59). Additionally, sections of the Methods (2.4) and Results (3.2) were removed.

Comment 2: I also suggest modifying the manuscript title to:
"Alterations in the Microbiota of Horses with FWS"

Response 2: We appreciate the Reviewer’s suggestion and have changed the title accordingly.

Comment 3: Finally, the aim of the study is not clearly defined in the introduction. It is important that one of the final lines of the introduction explicitly states: “The aim of this work was…”.

Response 3: We appreciate the Reviewer’s suggestion and have added a final paragraph to the expanded Introduction clearly stating the Aims of the study (lines 61 to 65).

Comment 4: Materials and methods

Line 76-77: it is necessary an reference to validate this method of diagnosis.

Response 4: We have added a reference and clarified the text (lines 70-73)

Comment 5: 2.7) Statistics: you need to indicate p value of significance

Response 5: We have clarified in the Methods that p < 0.05 was considered significant for traditional univariate and multivariate tests, while a p value adjusted for multiple tests was used in differential abundance testing (with the same threshold for type I error) (lines 140 to 145).

Comment 6: Conclusions

A conclusion stating that B. coli is not the causative agent of FWS, without providing justification for why this is an important conclusion, is a conclusion based on unfounded reasoning. The entire introduction suggested that this bacterium would not play a significant role in this pathology.

Response 6: As we have opted to remove the B. coli testing and data completely from the manuscript, the conclusion and all other sections have been revised to focus on the microbiota.

Reviewer 2 Report

Comments and Suggestions for Authors

Porter et al. analyzed the fecal microbiota of horses with fecal water syndrome and negative control horses. The goal was to determined whether Balantidium coli is associated with FWS or this condition is associated with intestinal dysbiosis. The diagnosis of B. coli was based on PCR with primers taken from a study of Tibetan sheep. The manuscript does not refer to any confirmatory experiments to ensure that the PCR actually amplifies B. coli DNA, nor is it stated whether the 18S or ITS primers described in [14] were used. Given the importance of an accurate B. coli diagnosis for this study, a simple confirmation that the amplicons actually came from B. coli by Sanger sequencing seems warranted. The 16S analyses also lack controls, like duplicated or triplicated amplicons from selected horses to assess biological and technical variation in the sequence data.

Fig. S1: Y axis is labeled “fold-change” whereas legend says mean copy number. Please specify whether the data represent raw abundance, concentration or change.

Fig. 3. Please state the meaning of the connecting lines.

Fig. S2A. Legend refers to lines, but no lines are shown.

Line 153. “Differential abundance” of what? ASVs?

Lines 199/213. Please specify the meaning of “health status”. On line 79 it says that horses were healthy. Does status refer to FWS?

Author Response

Comment 1: Porter et al. analyzed the fecal microbiota of horses with fecal water syndrome and negative control horses. The goal was to determine whether Balantidium coli is associated with FWS or this condition is associated with intestinal dysbiosis. The diagnosis of B. coli was based on PCR with primers taken from a study of Tibetan sheep. The manuscript does not refer to any confirmatory experiments to ensure that the PCR actually amplifies B. coli DNA, nor is it stated whether the 18S or ITS primers described in [14] were used. Given the importance of an accurate B. coli diagnosis for this study, a simple confirmation that the amplicons actually came from B. coli by Sanger sequencing seems warranted.

Response 1: We appreciate and agree with the Reviewer’s suggestion. Based on these concerns, concerns of another Reviewer, and the overall lack of significant findings regarding B. coli, we have opted to remove that portion of the study altogether and focus solely on the fecal microbiome in the context of FWS.

Comment 2: The 16S analyses also lack controls, like duplicated or triplicated amplicons from selected horses to assess biological and technical variation in the sequence data.

Response 2: While we have performed such studies in the past, the current studies were not designed to evaluate biological and technical variation, and we did not generate or sequence duplicate libraries from the same sample. Our laboratory (the University of Missouri Metagenomics Center) follows stringent SOPs and routinely runs positive (microbial community standards, ZymoBIOMICs) and negative (reagent) controls alongside experimental samples but did not do so in the current study. Respectfully, we are confident in our methodology and do not believe that the lack of such information detracts from the findings reported here.

Comment 3: Fig. S1: Y axis is labeled “fold-change” whereas legend says mean copy number. Please specify whether the data represent raw abundance, concentration or change.

Response 3: We apologize for the discrepancy. As we have opted to completely remove the B. coli portion of the study, this Figure is no longer included in the revised manuscript.

Comment 4: Fig. 3. Please state the meaning of the connecting lines.

Response 4: We apologize for the omission. The lines denote the distance (i.e., dissimilarity) between samples from horses on the same farm.  We have clarified this in the Figure legend for this figure (now Figure 2) in lines 186-187.

Comment 5: Fig. S2A. Legend refers to lines, but no lines are shown.

Response 5: We apologize for the error and have removed those words from the figure legend.

Comment 6: Line 153. “Differential abundance” of what? ASVs?

Response 6: Differential abundance testing was performed across all taxonomic levels from amplicon sequence variant (ASV) to phylum, and only significantly different features are reported.  This has been clarified in the revision (line 139).

Comment 7: Lines 199/213. Please specify the meaning of “health status”. On line 79 it says that horses were healthy. Does status refer to FWS?

Response 7: We appreciate the Reviewer’s careful reading and regret the wording in question. Our intent was to indicate that all horses, including FWS-affected and control horses, were ostensibly healthy aside from FWS, based on results of a thorough physical examination by the attending veterinarian. We have amended the text in question (line 76) to read “overall health (independent of FWS)”.

Round 2

Reviewer 1 Report

Comments and Suggestions for Authors

I would like to thank the authors for having taken all my suggestions and corrections into account.

Although the manuscript has improved substantially in terms of scientific rigor, I still have some concerns regarding its scientific impact.
In particular, the fact that the discussion section spans less than one page raises some concerns.

Would it be possible to expand this section further?
What are the influencing factors behind microbiota alterations, and can they be explained in more detail?
What is the impact of such alterations?
Most importantly, what future studies should be conducted, and how can the methodology be improved?

Author Response

I would like to thank the authors for having taken all my suggestions and corrections into account.

Although the manuscript has improved substantially in terms of scientific rigor, I still have some concerns regarding its scientific impact.
In particular, the fact that the discussion section spans less than one page raises some concerns.

Would it be possible to expand this section further?
What are the influencing factors behind microbiota alterations, and can they be explained in more detail?
What is the impact of such alterations?
Most importantly, what future studies should be conducted, and how can the methodology be improved?

Response: We appreciate the Reviewer’s continued efforts. The reduced length of the Discussion was due to the removal of portions related to Balantidium coli. That said, we agree with the Reviewer that the Discussion could be expanded. As such, we have added text describing other factors capable of influencing the microbiota, other possible reasons for discrepancies with other studies, and needs for future studies (lines 233-261).

Reviewer 2 Report

Comments and Suggestions for Authors

Porter et al. have responded to the reviewer’s comments by removing data pertaining to B. coli. The revised manuscript continues to suffer from several weaknesses. First, is the lack of controls. Without a minimum of replication and/or internal standards, it is difficult to assess how much noise and how much signal is in the data. Second, the analysis of the 16S sequences fails to consider several variables which may well impact the intestinal microbiota (Table S1). Given the relative similarity of the microbiota in afflicted and control horses, as shown in Fig. 2, the effect of farm (considered on line 176-180), age, “Current Ration”, “Current Medications” should also be investigated. This could be done by defining these variables as co-variates and assessing the impact of FWS status after the effect of additional variables has been subtracted. Alternatively, testing for the significance of variables which are known to impact the microbiota, such as ration, medication (NSAID, see Whitfield-Cargile et al., 2018, PLoS ONE) and perhaps age should be considered to strengthen the conclusions. The association between the main variables should perhaps also be tested.

Minor points:

Fig. 2 is in my view uninterpretable. Why not display two PCoA plot for each distance metric, one colored by FWS presence/absence and the other colored by farm?

Balantidium coli is still mentioned in keywords even though the analyses pertaining to this parasite were removed.

Table S1, the two worksheets seem to contain the same information. Sorted by Farm does not show farm information.

Author Response

Porter et al. have responded to the reviewer’s comments by removing data pertaining to B. coli. The revised manuscript continues to suffer from several weaknesses. First, is the lack of controls. Without a minimum of replication and/or internal standards, it is difficult to assess how much noise and how much signal is in the data. Second, the analysis of the 16S sequences fails to consider several variables which may well impact the intestinal microbiota (Table S1). Given the relative similarity of the microbiota in afflicted and control horses, as shown in Fig. 2, the effect of farm (considered on line 176-180), age, “Current Ration”, “Current Medications” should also be investigated. This could be done by defining these variables as co-variates and assessing the impact of FWS status after the effect of additional variables has been subtracted. Alternatively, testing for the significance of variables which are known to impact the microbiota, such as ration, medication (NSAID, see Whitfield-Cargile et al., 2018, PLoS ONE) and perhaps age should be considered to strengthen the conclusions. The association between the main variables should perhaps also be tested.

Response: While we appreciate the Reviewer’s suggestions concerning sample replicates, but no such testing was done with these samples, and they are no longer available. That being said, we would respectfully submit that the presence of multiple positive findings (e.g., differences in beta-diversity and relative abundance of multiple taxa) obviates those issues. While not run with these samples, our lab (the MU Metagenomics Center, MUMC) routinely processes mock community standards (ZymoBIOMICS D6300) alongside experimental samples as an internal control. Thus, we do not have additional samples or funding to allow sequencing of replicates, but we believe our methods are robust.

The Reviewer raises an excellent point regarding consideration of other factors, and we have expanded the analysis to include 1) associations between age or rations and FWS, and 2) the effect of farm, age, and FWS status on beta-diversity. Specifically, we first compared age of affected and control horses using a t-test and found a significant difference (p = 0.027). While this obviously does not implicate age in FWS (available control horses on the same farms just happened to be, on average, a few years younger than affected horses), it did lead us to include age as a factor in considerations of beta-diversity. To evaluate associations between different rations and FWS, chi-square analyses were performed for access to hay and pasture (all horses received grain). Neither hay (χ2 = 0.036, df = 1, p = 0.85) or pasture (χ2 = 0.005, df = 1, p = 0.945) were associated with FWS. We recognize a limitation of this approach is the lack of detail regarding proportions of each ration and include that in the Discussion (lines 255-259), but those data are not available. Information regarding current medications was considered but deemed sufficiently sparse to preclude its utility. To address the Reviewer’s suggestions regarding correction for other covariates in assessments of beta-diversity, we thus opted to correct for farm (previously identified as a significant effect) before evaluating the effect of FWS status and age. PERMANOVA were run using the formula [dist ~ farm + (group * age)]; this determines the variance associated with farm before assigning variance to group and age. Collectively, the results indicate that while Farm explains a greater amount of overall variance (based on R2 value and inflated due to the large number of farms), samples exhibited a stronger separation based on FWS status (F value of FWS status > Farm).  Age and group:age interactions did not have significant effects. This was consistent using weighted and unweighted dissimilarities. These findings have been added to the Results (lines 176-193) and are considered in the revised Discussion (lines 233-261). We sincerely appreciate the Reviewer’s suggestions and believe they have strengthened the reported findings and conclusions.

Minor points:

Fig. 2 is in my view uninterpretable. Why not display two PCoA plot for each distance metric, one colored by FWS presence/absence and the other colored by farm?

Response: Per the Reviewer’s suggestion, we have revised Figure 2 to include matched panels colored by presence/absence of FWS and by farm. 

Balantidium coli is still mentioned in keywords even though the analyses pertaining to this parasite were removed.

Response: With apologies, Balantidium coli has been removed from the Keywords.

Table S1, the two worksheets seem to contain the same information. Sorted by Farm does not show farm information.

Response: We have revised Table S1 to include ‘Farm’ as a factor.